# Post-traumatic stress disorder symptoms among healthcare workers during the COVID-19 pandemic: Analysis of the HERO Registry

Eli N. Rice[1]☯, Haolin Xu[2]☯, Ziyi Wang[2]☯, Laura Webb[2]☯, Laine Thomas[2]☯, Emilie F. Kadhim[3]☯, Julio C. Nunes[5]☯, Kathryn C. Adair[4]☯, Emily C. O'Brien[2,6]☯ *

1 Department of Psychology, University of Pittsburgh, Pittsburgh, PA, United States of America, 2 Duke Clinical Research Institute, Duke University School of Medicine, Durham, NC, United States of America, 3 Social & Behavioural Health Sciences Division, Dalla Lana School of Public Health, University of Toronto, Toronto, ON, Canada, 4 Duke Center for Healthcare Safety and Quality, Duke University Health System, Durham, NC, United States of America, 5 Department of Psychiatry, Yale University School of Medicine, New Haven, CT, United States of America, 6 Department of Population Health Sciences, Duke University, Durham, NC, United States of America

☯ These authors contributed equally to this work.
* emily.obrien@duke.edu

**Data Availability Statement:** Investigators interested in accessing the HERO data for analysis may submit a proposal to the HERO Publications

## Abstract

Little is known about the mental health consequences of the COVID-19 pandemic in healthcare workers (HCWs). Past literature has shown that chronic strain caused by pandemics can adversely impact a variety of mental health outcomes in HCWs. There is growing recognition of the risk of stress and loss of resilience to HCWs during the COVID-19 pandemic, although the risk of post-traumatic stress disorder (PTSD) symptoms in HCWs during the COVID-19 pandemic remains poorly understood. We wanted to understand the relationship between the COVID-19 pandemic and the risk of PTDS symptoms in HCWs during the COVID-19 pandemic. We surveyed 2038 health care workers enrolled in the Healthcare Worker Exposure Response & Outcomes (HERO) study, which is a large standardized national registry of health care workers. Participants answered questions about demographics, COVID-19 exposure, job burnout, and PTSD symptoms. We characterize the burden of PTSD symptoms among HCWs, and determined the association between high PTSD symptoms and race, gender, professional role, work setting, and geographic region using multivariable regression. In a fully adjusted model, we found that older HCWs were less likely to report high PTSD symptoms compared with younger HCWs. Additionally, we found that physicians were less likely to report high PTSD symptoms compared with nurses. These data add to the growing literature on increased risks of mental health challenges to healthcare workers during the COVID-19 pandemic.

## Introduction

The emergence of the severe acute respiratory syndrome coronavirus 2 (SARS-CoV-2) and COVID-19 placed an unprecedented demand on the healthcare system in 2020. Now, over

Committee for review (https://heroesresearch.org/projects_pubs/).

**Funding:** The HERO Registry is funded through a Patient-Centered Outcomes Research Institute Award (COVID-19-2020-001). The program is coordinated by the Duke Clinical Research Institute and leverages PCORnet®, the National Patient-Centered Clinical Research Network. The funders had no role in study design, data collection and analysis, decision to publish, or preparation of the manuscript.

**Competing interests:** The authors have declared that no competing interest exist.

two years later, with multiple new viral strains and a growing burden of burnout among health system workers (HCWs), key questions remain about the mental health consequences of exposure to acute and chronic stressors. Post-traumatic stress disorder (PTSD) is a disorder that develops in some people who have experienced a shocking, scary, or dangerous event [1]. Common examples of harmful experiences that may result in PTSD include military combat, natural disasters, and violent assault. Prior research indicates that 'helpers' during human disasters often face psychological trauma [2]. Analyses of post-9/11 experiences among first responders underscored the psychological risks for professionals exposed to suffering [3], with terms like compassion fatigue, secondary traumatic stress, and vicarious trauma used to describe such effects. This suggests that healthcare providers treating COVID-19 patients may also experience similar trauma, given their critical role in response to this crisis. Furthermore, recent data has shown that the chronic strain caused by pandemics has a psychological effect on a variety of health outcomes in HCWs, including burnout, depression, and anxiety [4]. The impacts of COVID-19 and other recent health system stressors on the rates and risk factors for post-traumatic stress disorder (PTSD) symptoms remain poorly understood [5–7].

Past studies in other pandemic settings, including the severe acute respiratory syndrome (SARS) outbreak in 2003, reported increased PTSD symptoms among HCWs [8, 9]. In the setting of COVID-19, factors associated with PTSD are less well-understood. Early studies found that nurses, often at the front line of the pandemic in high-risk settings, experienced increased psychological distress [10]. Additionally, female healthcare workers have higher rates of burnout and distress experiences [11, 12], and HCWs from underrepresented racial and ethnic groups have been disproportionately impacted by the mental health burden of the COVID-19 pandemic [13, 14]. Moral injury, the "psychological, biological, spiritual, behavioral and social impact of perpetrating, failing to prevent, or bearing witness to acts that transgress deeply held moral beliefs and expectations" [15], has been associated with increased risk of PTSD and symptoms of PTSD, suicidality, and substance abuse [16]. Healthcare workers during the COVID-19 setting have experienced high levels of moral injury, comparable to combat veterans in the post-9/11 era [17]. Finally, the geography of the work setting itself may be an important determinant of post-traumatic symptoms, with geographic variation in case burden and available resources to support healthcare workers in different geographic regions [18].

Further understanding of the mental health sequelae of pandemic-related trauma is critical to implementing interventions to promote HCWs retention, improve their quality of life, and reduce the long-term health effects of high stress settings. We used a previously validated PTSD questionnaire [19] and the Healthcare Worker Exposure Response & Outcomes (HERO) registry, a national registry created to understand the impact of COVID-19 on healthcare workers, to (a) characterize the prevalence of PTSD symptoms during the COVID-19 pandemic among health care workers in the HERO registry, and (b) estimate the association between HCW characteristics (race, gender, professional role, work setting, and geographic region). Given the relationship between prior pandemics and the development of PTSD symptoms, we hypothesized that we would observe elevated levels of PTSD symptoms in HCWs working during the COVID-19 pandemic and that PTSD symptoms would be associated with specific HCW characteristics such as age, gender, and type of professional role.

## Methods

### Hero Registry

The HERO research program (*NCT04342806)* was created in 2020 in response to the COVID-19 pandemic as an initiative of the Patient Centered Outcomes Research Institute. Recruitment was virtual and open to all individuals who wanted to join, however, the Registry used HCW-

focused recruitment strategies within academic medical centers. All study procedures were reviewed by the Duke University School of Medicine Institutional Review Board and approved by the Western Institutional Review Board (Pro00105284).

The Registry was a fully remote online study that captured perspectives through ongoing virtual surveys and return of results. Although participation in the HERO Registry was through an online portal, enrollment and engagement for the Registry supported by the National Patient-Centered Clinical Research Network (PCORnet) infrastructure and 34 PCORnet sites. The study primarily enrolled participants through marketing campaigns and word of mouth, but also through targeted recruitment efforts at PCORnet sites participating in one of the research projects leveraging the Registry (HERO-HCQ or HERO-TOGETHER) and that used the Registry for pre-screening for project eligibility. Participants self-enrolled and responded to surveys through an online portal that was supported by desktop and mobile applications, as such we were unable to capture metrics such as response rate.

The overall HERO study population consisted of healthcare workers, their family members, and members of the community. For this current analysis, inclusion criteria included: $\geq 18$ years old, able to read English or Spanish, and were a HCW. The registry defined a HCW as anyone that worked in a facility that provides healthcare services to patients. Once deemed eligible, participants completed a brief screening to confirm eligibility on the study website (heroresearch.org) and signed an electronic consent form. Participants who consented and entered a valid email address were sent an invite through email and were given the opportunity to participate in the PTSD survey during the week between 4/29/2022 and 5/6/2022. By querying HERO data, 51464 participants consented between April and May and 3593 (7%) responded to the PTSD hot topics survey. Participants created a profile with contact information, demographics, employment characteristics (hours worked in the past week, work settings, professional role, etc.), and interest in participating in future studies. Since the study was online, no enrollment cap was set. Participants then answered periodic survey questions about COVID-19 exposures, histories, and testing, along with questions about job burnout.

## Survey tool

HERO participants were invited to contribute to the PTSD survey and submitted responses between April and May 2022. This survey included a 5 question Diagnostic and Statistical Manual of Mental Disorders (DSM)-5 PTSD screening survey which has been previously adapted and validated to screen for PTSD symptoms in COVID-19 patients (Table 1) [19]. Although modified and validated, this survey was not intended to represent formal PTSD diagnosis but rather used as a screening survey to screen for potential PTSD symptoms. This survey altered the questions to replace the phrase "the event" with "COVID-19." For example, the original questionnaire asks if the respondent has "felt guilty or unable to stop blaming yourself or others for any problems the event may have caused" compared with the revised questionnaire which asks if the respondent has "felt guilty or unable to stop blaming yourself or others

**Table 1. Adapted DSM-5 PTSD questionnaire.**

| Items "In the past month, have you . . ." |
| --- |
| Q1.. . . had nightmares about COVID-19 or thought about COVID-19 when you did not want to? |
| Q2.. . . tried hard not to think about COVID-19, or gone out of your way to avoid situations that reminded you of COVID-19? |
| Q3.. . . been constantly on guard, watchful, or easily startled? |
| Q4.. . . felt numb or detached from people, activities, or your surroundings? |
| Q5.. . . felt guilty or unable to stop blaming yourself or others for any problems COVID-19 may have caused? |

for any problems COVID-19 may have caused." The response options were identical to those in the original questionnaire (1 = Yes, 0 = No). As recent literature has shown that PTSD is congruent with an overall score of 3 or higher points of PTSD symptoms using screening tools [20], we used this threshold as the cutoff to indicate high levels of PTSD symptoms in our cohort.

**Statistical analysis**

Our main objective was to describe the rate of high levels of self-reported PTSD symptoms in HCWs and to characterize variation in the rate of high PTSD symptoms across populations. We defined the rate of PTSD symptoms as high/low derived from a 5-point symptom scale, in which a scale of >3 symptoms reported were defined as "high." We described the rate of high PTSD symptoms in different HCW cohorts, including age, gender, race/ethnicity, as well as the subject's work setting (type of facility, region of facility and professional role), and work hours.

Our second objective was to assess the unadjusted and adjusted associations between each of the variables of interest with the rate of high PTSD symptoms using logistic regressions. For this model, we similarly used a binary outcome to define the rate of PTSD symptoms as high/low derived from a 5-point symptom scale, in which a scale of ≥3 symptoms reported were defined as "high." Research in a large sample of VA primary care patients found that a cut-point of 4 ideally balanced false negatives and false positives for the overall sample and for men. However, for women, a cut-point of 4 resulted in high numbers of false negatives. Practitioners may consider a lower cut-point for women in some settings if evaluation resources are available [21]. Since our sample is predominantly female (>80%) we chose a PTSD score of 3 or higher to indicate PTSD symptoms. The explanatory variables considered were age, gender, region, race and ethnicity, healthcare worker role, and the number of work hours in the past week.

**Results**

A total of 2038 healthcare professionals enrolled in the HERO study completed the PTSD Survey (Table 2). The majority of HCWs were nurses (27%) followed by other health care workers (18%), diagnosing and treating practitioners (15%), and physicians (14%). The majority of participants were White non-Hispanic (81%) in the 30–49 age range (60%). Most HCW reported working an average of 40–59 hours per week (46%) in a hospital setting (54%), and the majority were located in the southern US (41%).

The frequencies of responses to individual items are shown in Table 3. Of participants with complete survey responses, individuals endorsing 1 or more items was the most frequent, with a consistent and gradual decrease to individuals selecting 2 or more, 3 or more, 4 or more, and 5 items.

Using a chi-square test, we found that age and gender are significantly associated with PTSD symptoms (Table 4), (p < .01, p < .01 respectively).

To explore the relationship between participant characteristics and PTSD symptoms, we ran a logistic regression analysis with high PTSD symptom score (3 or more items) as the outcome of interest (Table 5). In unadjusted models, females had higher odds of high PTSD symptoms compared with males [odds ratios (OR), 1.6, (1.20, 2.15). HCWs in the two oldest age groups (50–64 years old and 65+ years old) were less likely to have high PTSD symptoms compared with those in the 30–49 year old group [OR .71, (.55, .92), OR .31, (.12, .78), respectively]. Additionally, physicians and paramedic/emergency medical technicians were less likely to have high PTSD symptoms compared with nurses [OR .59, (.42, .84), OR .43, (.19, .97), respectively].

**Table 2. Demographics of HCWs.**

| Variable | Level | Overall (N = 2038) | |
|---|---|---|---|
| DEMOGRAPHICS | | Total (n) | Value/Percentage |
| Age | Median (25th– 75th percentile) | 2038 | 40.0 (33.0–51.0) |
| Age (years) | 65+ | 55 | 2.7 |
| | 50–64 | 510 | 25.0 |
| | 30–49 | 1213 | 59.5 |
| | 18–29 | 260 | 12.8 |
| Are you identified as female? | Yes | 1669 | 81.9 |
| | No | 369 | 18.1 |
| Race/Ethnicity, n (%) | Other | 64 | 3.1 |
| | Asian | 103 | 5.1 |
| | Hispanic, any race | 128 | 6.3 |
| | Black, non-Hispanic | 90 | 4.4 |
| | White, non-Hispanic | 1653 | 81.1 |
| Geographic Region | Region 4 West | 218 | 10.7 |
| | Region 3 South | 844 | 41.4 |
| | Region 2 Midwest | 474 | 23.3 |
| | Region 1 Northeast | 502 | 24.6 |
| What is your primary role in the healthcare setting where you work? | Healthcare support, administrative, and research staff [D] | 290 | 14.2 |
| | Health technologists, technicians, and clinical support staff [D] | 59 | 2.9 |
| | Other Health Diagnosing and Treating Practitioners [D] | 312 | 15.3 |
| | Physician's assistant/Nurse practitioner (PA/NP) [D] | 124 | 6.1 |
| | Other (free text) [D] | 358 | 17.6 |
| | Paramedic/ Emergency Medical Technician [D] | 51 | 2.5 |
| | Nurse (RN/LPN) [D] | 550 | 27.0 |
| | Physician | 294 | 14.4 |
| In what sort of healthcare facility do you work? | Other | 510 | 25.0 |
| | Urgent care clinic/Emergency Services | 61 | 3.0 |
| | Outpatient Clinic/Facility | 307 | 15.1 |
| | Skilled Nursing Facility/Nursing Facility(SNF/NF) | 68 | 3.3 |
| | Hospital | 1092 | 53.6 |
| How many hours did you work in a healthcare setting in the past week? | 60+ hrs | 117 | 5.7 |
| | 40–59 hrs | 936 | 45.9 |
| | 20–39 hrs | 612 | 30.0 |
| | 1–19 hrs | 239 | 11.7 |
| | 0 hr | 134 | 6.6 |

In fully adjusted models, the association between gender and high PTSD symptoms was attenuated and no longer significant (p = .09). The association between older age and high PTSD symptoms persisted [50–64 years old: OR 0.69, (.53, .90), 65+ years: OR 0.36, (.14, .92),]. In fully adjusted models, physicians were less likely to report high PTSD symptoms compared with nurses [OR .64, (.44, .93)]. Working longer hours (60+ hours per week) was also associated with greater odds of high PTSD symptoms [OR 1.59, (1.02, 2.47)] compared to working 40–59 hours a week.

## Discussion

The COVID-19 pandemic has led to high levels of psychological distress among the general population, with especially notable impacts on frontline healthcare workers. In this large

**Table 3. PTSD survey responses in HERO HCW participants (n = 2038).**

| Label | n | N | % |
|---|---|---|---|
| PTSD Questions: In the past month, have you . . . | | | |
| Q1.. . . had nightmares about COVID-19 or thought about COVID-19 when you did not want to? | 627 | 2038 | 30.8% |
| Q2.. . . tried hard not to think about COVID-19, or gone out of your way to avoid situations that reminded you of COVID-19? | 687 | 2035 | 33.8% |
| Q3.. . . been constantly on guard, watchful, or easily startled? | 462 | 2035 | 22.7% |
| Q4.. . . felt numb or detached from people, activities, or your surroundings? | 746 | 2032 | 36.7% |
| Q5.. . . felt guilty or unable to stop blaming yourself or others for any problems COVID-19 may have caused? | 292 | 2031 | 14.4% |
| Indicated 0 items | 837 | 2038 | 41.1 |
| Indicated 1 items | 411 | 2038 | 20.2 |
| Indicated 2 items | 313 | 2038 | 15.4 |
| Indicated 3 items | 233 | 2038 | 11.4 |
| Indicated 4 items | 142 | 2038 | 7.0 |
| Indicated 5 items | 102 | 2038 | 5.00 |
| Indicated 1 or more items [D] | 1201 | 2038 | 58.9% |
| Indicated 2 or more items [D] | 790 | 2038 | 38.8% |
| Indicated 3 or more items [D] | 477 | 2038 | 23.4% |
| Indicated 4 or more items [D] | 244 | 2038 | 12.0% |
| Indicated 5 items | 102 | 2038 | 5.0% |

national survey of healthcare workers caring for patients during the COVID-19 pandemic, we found that PTSD symptoms present in healthcare workers, with highest prevalence rates among nursing staff. Using multivariable analysis, we found that younger age, nursing role, and 60+ hour work weeks were strongly associated with a greater rate of high levels of PTSD symptoms. These data add to the growing literature on the risks to healthcare workers of excessive stress during the ongoing COVID-19 pandemic [22].

Our study confirms most existing literature which examines gender disparities in psychological outcomes during the pandemic. Prior studies have reported greater vulnerability among female healthcare workers to mental health disorders such as PTSD than their male counterparts [11, 23]. Although we found higher unadjusted odds of PTSD symptoms among women compared with men, this association was attenuated after adjusting for other factors, including professional role. These results suggest that at least part of the association between female gender and adverse psychological sequelae may be due to women holding professional roles that are more vulnerable to PTSD symptoms, such as nursing. More research is needed into possible interactions between gender and professional roles and how to best support HCWs with multiple risk factors for work-related PTSD.

Consistent with previous literature, we saw that nurses had greater risk of high PTSD symptoms compared with physicians [12]. We surmise that this association could be due to nurses, particularly those having the close proximity to critically ill patients, are at high risk for COVID-19 infection and transmission [24–26]. Past literature has shown that ICU nurses have higher levels of stress and PTSD symptoms compared to non-ICU nurses [27]. The COVID-19 pandemic has not only put nurses on the frontline but has increased hospital system stressors, including demand for personal protective equipment, inpatient and ICU beds, and healthcare workers. Past literature has shown that when healthcare providers are in situations where the demand exceeds the available resources, stress often follows [28].

**Table 4. Demographics and PTSD results of HCWs.**

| Variable | Level | PTSD (> = 3) (N = 477) | | No PTSD (<3) (N = 1561) | | P-value + | % Std. Diff. |
|---|---|---|---|---|---|---|---|
| **Age** | **Median** (25th– 75th percentile) | **477** | **38.0 (32.0–48.0)** | **1561** | **41.0 (33.0–52.0)** | **< .0001** | **22.9** |
| Demographics | | Total | % | Total | % | | |
| Age | 65+ | 5 | 9.1 | 50 | 90.9 | 0.0002 | 24.2 |
| | 50–64 | 96 | 18.8 | 414 | 81.2 | | |
| | 30–49 | 298 | 24.6 | 915 | 75.4 | | |
| | 18–29 | 78 | 30.0 | 182 | 70.0 | | |
| Are you identified as female? | Yes | 414 | 24.8 | 1255 | 75.2 | 0.0015 | 17.3 |
| | No | 63 | 17.1 | 306 | 82.9 | | |
| Race/Ethnicity, n (%) | Other | 18 | 28.1 | 46 | 71.9 | 0.25 | 11.8 |
| | Asian | 20 | 19.4 | 83 | 80.6 | | |
| | Hispanic, any race | 33 | 25.8 | 95 | 74.2 | | |
| | Black, non-Hispanic | 28 | 31.1 | 62 | 68.9 | | |
| | White, non-Hispanic | 378 | 22.9 | 1275 | 77.1 | | |
| Geographic Region [D] | Region 4 West | 59 | 27.1 | 159 | 72.9 | 0.57 | 7.3 |
| | Region 3 South | 190 | 22.5 | 654 | 77.5 | | |
| | Region 2 Midwest | 110 | 23.2 | 364 | 76.8 | | |
| | Region 1 Northeast | 118 | 23.5 | 384 | 76.5 | | |
| EMPLOYMENT | | | | | | | |
| What is your primary role in the healthcare setting where you work? | Healthcare support, administrative, and research staff [D] | 72 | 24.8 | 218 | 75.2 | 0.1 | 19.3 |
| | Health technologists, technicians, and clinical support staff [D] | 11 | 18.6 | 48 | 81.4 | | |
| | Other Health Diagnosing and Treating Practitioners [D] | 71 | 22.8 | 241 | 77.2 | | |
| | Physician's assistant/Nurse practitioner (PA/NP) [D] | 30 | 24.2 | 94 | 75.8 | | |
| | Other (free text) [D] | 84 | 23.5 | 274 | 76.5 | | |
| | Paramedic/Emergency Medical Technician [D] | 7 | 13.7 | 44 | 86.3 | | |
| | Nurse (RN/LPN) [D] | 149 | 27.1 | 401 | 72.9 | | |
| | Physician | 53 | 18.0 | 241 | 82.0 | | |
| In what sort of healthcare facility do you work? | Other | 113 | 22.2 | 397 | 77.8 | 0.4 | 10.6 |
| | Urgent care clinic/Emergency Services | 13 | 21.3 | 48 | 78.7 | | |
| | Outpatient Clinic/Facility | 63 | 20.5 | 244 | 79.5 | | |
| | Skilled Nursing Facility/Nursing Facility(SNF/NF) | 14 | 20.6 | 54 | 79.4 | | |
| | Hospital | 274 | 25.1 | 818 | 74.9 | | |
| How many hours did you work in a healthcare setting in the past week? | 60+ hrs | 34 | 29.1 | 83 | 70.9 | 0.6 | 8.2 |
| | 40–59 hrs | 213 | 22.8 | 723 | 77.2 | | |
| | 20–39 hrs | 144 | 23.5 | 468 | 76.5 | | |
| | 1–19 hrs | 53 | 22.2 | 186 | 77.8 | | |
| | 0 hr | 33 | 24.6 | 101 | 75.4 | | |

We also found that age is closely related to rates of PTSD symptoms, with younger workers having higher rates of PTSD symptoms than older workers. This is congruent with previous literature on risk factors for pandemic driven PTSD in HCWs [29]. Our data shows that older age may be protective against the risk of developing high levels of PTSD symptoms. This is

**Table 5. Associations of PTSD (score > = 3 vs <3) and HCW characteristics.**

| Parameter | Level | Unadjusted Analysis | | Adjusted Analysis | |
|---|---|---|---|---|---|
| | | OR (95% CI) | P-value | OR (95% CI) | P-value |
| Female | 1 = Yes | 1.60 (1.20, 2.15) | 0.0016 | 1.31 (0.95, 1.80) | 0.0959 |
| | 0 = No | Reference | | Reference | |
| Age | 0 = 18–29 | 1.32 (0.98, 1.77) | 0.0688 | 1.31 (0.97, 1.78) | 0.0781 |
| | 2 = 50–64 | 0.71 (0.55, 0.92) | 0.0098 | 0.69 (0.53, 0.90) | 0.0062 |
| | 3 = 65+ | 0.31 (0.12, 0.78) | 0.0127 | 0.36 (0.14, 0.92) | 0.0325 |
| | 1 = 30–49 | Reference | | Reference | |
| Race/Ethnicity | 2 = Black/African-American | 1.52 (0.96, 2.41) | 0.0734 | 1.58 (0.98, 2.55) | 0.0601 |
| | 3 = Hispanic/Latino (any race) | 1.17 (0.78, 1.77) | 0.4514 | 1.06 (0.69, 1.62) | 0.7819 |
| | 4 = Asian/Pacific Islander | 0.81 (0.49, 1.34) | 0.4179 | 0.81 (0.48, 1.35) | 0.4087 |
| | 5 = Other/mixed/PNTA | 1.32 (0.76, 2.30) | 0.3287 | 1.36 (0.77, 2.40) | 0.2822 |
| | 1 = White | Reference | | Reference | |
| Region | 2 = Region 2 Midwest | 0.98 (0.73, 1.32) | 0.9121 | 0.94 (0.69, 1.27) | 0.6899 |
| | 3 = Region 3 South | 0.95 (0.73, 1.23) | 0.6746 | 0.92 (0.70, 1.20) | 0.5271 |
| | 4 = Region 4 West | 1.21 (0.84, 1.74) | 0.3087 | 1.20 (0.83, 1.74) | 0.3234 |
| | 1 = Region 1 Northeast | Reference | | Reference | |
| Healthcare Role | 2 = Physician | 0.59 (0.42, 0.84) | 0.0035 | 0.64 (0.44, 0.93) | 0.0203 |
| | 93 = Paramedic/Emergency Medical Technician [D] | 0.43 (0.19, 0.97) | 0.0425 | 0.43 (0.19, 1.01) | 0.0541 |
| | 95 = Other (free text) [D] | 0.83 (0.61, 1.12) | 0.2217 | 0.81 (0.59, 1.11) | 0.1819 |
| | 96 = Physician's assistant/Nurse practitioner (PA/NP) [D] | 0.86 (0.55, 1.35) | 0.5096 | 0.84 (0.53, 1.33) | 0.4644 |
| | 97 = Other Health Diagnosing and Treating Practitioners [D] | 0.79 (0.57, 1.10) | 0.1612 | 0.74 (0.53, 1.04) | 0.0794 |
| | 98 = Health technologists, technicians, and clinical support staff [D] | 0.62 (0.31, 1.22) | 0.1646 | 0.65 (0.33, 1.30) | 0.2255 |
| | 99 = Healthcare support, administrative, and research staff [D] | 0.89 (0.64, 1.23) | 0.4789 | 0.83 (0.59, 1.17) | 0.2842 |
| | 91 = Nurse (RN/LPN) [D] | Reference | | Reference | |
| Work Hour | 0 = 0 hr | 1.11 (0.73, 1.69) | 0.6304 | 1.13 (0.74, 1.75) | 0.5718 |
| | 1 = 1–19 hrs | 0.97 (0.69, 1.36) | 0.8482 | 0.93 (0.65, 1.32) | 0.6777 |
| | 2 = 20–39 hrs | 1.04 (0.82, 1.33) | 0.7241 | 0.96 (0.75, 1.23) | 0.7445 |
| | 4 = 60+ hrs | 1.39 (0.91, 2.13) | 0.1306 | 1.59 (1.02, 2.47) | 0.0410 |
| | 3 = 40–59 hrs | Reference | | Reference | |

consistent with past literature concerning PTSD and veterans in which older individuals have lower rates of PTSD compared to younger individuals [30]. It is not quite clear why PTSD is more frequently reported among younger veterans, however, some have theorized that older individuals may be less likely to attribute their PTSD symptoms to psychological distress and more likely to attribute their symptoms to general aging [31]. We believe that our findings align with this possible explanation as younger HCWs are displaying higher levels of PTSD symptoms compared to older ones. Further research will need to be conducted to understand the relationship between age and PTSD symptoms among HCWs during pandemics.

As the COVID-19 pandemic continues to evolve from an acute crisis into a chronic presence in our healthcare system, more work is needed to understand the long-term risks posed by working in a chronic pandemic environment. Past research has shown that high PTSD symptoms were found among 40% of hospital employees three years after the SARS outbreak [9]. In particular, little is known about the duration of PTSD symptoms following the COVID-19 pandemic, factors associated with resilience and full recovery from PTSD, whether COVID-19 related PTSD is associated with greater medication use or use of healthcare services, and what interventions might be most effective to prevent PTSD in future times of

healthcare system strain. Further work in these areas will support allocation of resources to address the current epidemic of burnout in the healthcare workforce.

Our study has several limitations, including the selected nature of the sample. HERO is a voluntary registry that may not be representative of all healthcare workers. We allowed all HCWs meeting inclusion criteria to enroll in our study to enhance generalizability, but survey data is self-reported and therefore subject to information bias. Moreover, although our adjusted model controlled for several important confounders such as gender, we cannot rule out the possibility of residual confounding from additional unmeasured factors. Also, due to the broad open-ended invitation of the study, we were unable to collect metrics on response rates. Additionally, we defined our region variable by North, South, East, and West, but did not base our analysis on rural vs. urban areas. Additionally, healthcare centers may differ in staff benefits including access to mental health resources. Furthermore, we did not measure information about prior PTSD diagnosis or treatment and used a brief scale to measure PTSD symptomatology. A more in-depth psychometric scale, including items about anxiety, depression, burnout along with PTSD would give us a greater idea of true PTSD scores. Lastly, our population included people with access to the online platform, which may limit generalizability to non-internet users.

Although out study has limitations, we also wanted to highlight the strengths. We collected data from a large cohort, with over two thousand participants from a variety of healthcare settings. Additionally, this study is novel as the topic of PTSD symptoms during the COVID-19 pandemic is not yet fully explored. We believe our study is a good starting point, demonstrating that COVID-19 is consistent with findings from previous studies about the prevalence of PTSD symptoms in prior pandemics. We urge further researchers to explore this issue in greater depth.

## Conclusion

In conclusion, we found that COVID-19 related psychological distress and PTSD symptoms are prevalent among healthcare workers. One's role within a healthcare context and age impact the likelihood of reporting high PTSD symptoms. We recommend that healthcare systems implement adequate and accessible mental health resources for healthcare workers. Furthermore, we suggest that further research needs to be conducted to see if there are preventative measures or interventions to put in place for future pandemics.

## Author Contributions

**Conceptualization:** Eli N. Rice, Ziyi Wang, Emily C. O'Brien.

**Data curation:** Eli N. Rice, Haolin Xu, Ziyi Wang, Emily C. O'Brien.

**Formal analysis:** Eli N. Rice, Haolin Xu, Ziyi Wang, Laura Webb, Laine Thomas, Kathryn C. Adair, Emily C. O'Brien.

**Investigation:** Julio C. Nunes, Emily C. O'Brien.

**Methodology:** Eli N. Rice, Laine Thomas, Emily C. O'Brien.

**Project administration:** Eli N. Rice, Emily C. O'Brien.

**Resources:** Kathryn C. Adair.

**Supervision:** Eli N. Rice, Kathryn C. Adair, Emily C. O'Brien.

**Visualization:** Eli N. Rice, Emily C. O'Brien.

**Writing – original draft:** Eli N. Rice, Haolin Xu, Ziyi Wang, Laura Webb, Laine Thomas, Emilie F. Kadhim, Julio C. Nunes, Emily C. O'Brien.

**Writing – review & editing:** Kathryn C. Adair.

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
