## [Decision Letter · Decision Letter 0]

18 Jun 2023

PONE-D-23-06322Post-Traumatic Stress Disorder symptoms among healthcare workers during the COVID-19 pandemic: Analysis of the HERO registryPLOS ONE

Dear Dr. Rice,

Thank you for submitting your manuscript to PLOS ONE. After careful consideration, we feel that it has merit but does not fully meet PLOS ONE’s publication criteria as it currently stands. Therefore, we invite you to submit a revised version of the manuscript that addresses the points raised during the review process.

We look forward to receiving your revised manuscript.

Kind regards,

Vincenzo De Luca

Academic Editor

PLOS ONE

“The HERO Registry is funded through a Patient-Centered Outcomes Research Institute Award (COVID-19-2020-001). The program is coordinated by the Duke Clinical Research Institute and leverages PCORnet®, the National Patient-Centered Clinical Research Network.”

Reviewers' comments:

Reviewer's Responses to Questions

**Comments to the Author**

1. Is the manuscript technically sound, and do the data support the conclusions?

Reviewer #1: Yes

Reviewer #2: Yes

2. Has the statistical analysis been performed appropriately and rigorously? 

Reviewer #1: I Don't Know

Reviewer #2: Yes

3. Have the authors made all data underlying the findings in their manuscript fully available?

Reviewer #1: No

Reviewer #2: No

4. Is the manuscript presented in an intelligible fashion and written in standard English?

Reviewer #1: Yes

Reviewer #2: Yes

5. Review Comments to the Author

Reviewer #1: 24 April 2023

Thank you for the opportunity to review the manuscript “Post- Traumatic Stress Disorder symptoms among healthcare workers during the COVID-19 pandemic: Analysis of the HERO registry”. It was interesting to go through the manuscript and to know that this was an outcome of the Patient Centered Outcome Research initiatives. The authors attempt to assess PTSD symptoms among healthcare workers during the COVID-19 pandemic and investigate association with demographic and other factors. Overall, the manuscript is well- written.

However, I have following comments in regard to various sections of the manuscript, and I suggest the authors to consider them in revising their paper. Please consider them as positive feedback and note that I am not an expert in statistical methods.

Title: The authors could reflect on the design of the study in the title itself besides mentioning the registry analysis.

Abstract: Please consider removing the exact values in odd ratios. Providing statements of your findings should suffice in the results.

Introduction: I suggest the authors to give a brief overview on why they think COVID-19 can be equated with Trauma definition for PTSD before proceeding on to literature depicting relation between PTSD and COVID-19.

The authors mention two objectives. When they say burden of PTSD symptoms, what do they mean? It is understandable that persistent stressor like COVID-19 will lead to an increase in levels of stress for most populations, including HCWs. How do they estimate that these symptoms are PTSD symptoms and not anxiety and/ or depression? When they say elevated level of PTSD symptoms, what is the baseline against which they measure the symptoms?

Methods:

What were the eligibility criteria except for being HCWs? Were there any duration criteria for symptoms mentioned?

I understand this was a voluntary registry- based study. However, Regarding sampling…was there any justification for sample size? How many invitations were sent? What was the response rate? What about missing data? Were there any outliers? Also, mention on the design of the study.

The authors mentioned cohort in one instance consisting of HCWs, their family members and members of the community. What was the relevance of this and why is it termed as cohort? Please clarify.

Survey tools

DSM- 5 PTSD screening survey: I suggest the authors to briefly mention regarding this instrument and list its psychometric properties in the given context. The questions in this survey appear vague and threshold of 3 could be met by any depressed or anxious individual as well. Please comment on this statement.

Statistical analysis

I suggest the authors to comment on the distribution of the variables of interest. I see that median has been reported for age, and I assume that it is skewed. Please provide details regarding other variables. Also, please cite the reference for PTSD threshold as 3 and above. Regarding unadjusted model, which statistical test was used? Were assumptions met for multivariate logistic regression? Please provide rationale for inclusion of variables in adjusted models.

Discussion

Discussion needs rewriting. In regard to first objective, how do the authors conclude that there is high level of PTSD symptoms in all levels of HCWs. Please justify.

In line 177- 178, the authors conclude with the statement that excessive stress and loss of resilience occurs to the HCWs during the pandemic. Please clarify on how raised PTSD symptoms means loss of resilience.

Discussion on the association between gender and PTSD is not enough since there is abundant literature on this. Most of the discussion appears generic on relation between psychological sequalae and gender. Please be more specific to in relation to PTSD. Same goes for association with nurses compared to physicians. With age, the evidence is mixed. Please discuss in light of those findings.

Limitations

The authors have listed few limitations. I believe this study has more limitations and biases such as using only screening questionnaire, not ruling out depression or anxiety or even burnout, which are quite common in this scenario. There could be many other factors at play which have not been considered in the given study.

I also suggest the authors to highlight the strengths of this study despite the limitations.

I wish the authors all the best.

Reviewer #2: The article is technically sound but the results should be discussed more thorughly, taking into account other possible explanations of the link between PTSD and demographic variables. It is needed to take into account that other studies, conducted in general population, also revealed a higher level of distress among women during the pandemic (e.g. https://iaap-journals.onlinelibrary.wiley.com/doi/abs/10.1111/aphw.12234 ). It would be also worth taking into account the previous studies concerning the link between age and distress during the COVID-19 pandemic, e.g. https://pubmed.ncbi.nlm.nih.gov/33388494/ Also, as far as I understand, the study was conducted mostly among inhabitants of USA. Does the country and culture play any role here? If so, why? I think that the authors should include the cross-cultural perspective in the discussion.

6. PLOS authors have the option to publish the peer review history of their article (what does this mean?). If published, this will include your full peer review and any attached files.

Reviewer #1: **Yes: **Saraswati Dhungana

Reviewer #2: No

---

## [Author Response · Author response to Decision Letter 0]

28 Jul 2023

REVIEWER #1

Reviewer: Thank you for the opportunity to review the manuscript “Post- Traumatic Stress Disorder symptoms among healthcare workers during the COVID-19 pandemic: Analysis of the HERO registry”. It was interesting to go through the manuscript and to know that this was an outcome of the Patient Centered Outcome Research initiatives. The authors attempt to assess PTSD symptoms among healthcare workers during the COVID-19 pandemic and investigate association with demographic and other factors. Overall, the manuscript is well- written.

However, I have following comments in regard to various sections of the manuscript, and I suggest the authors to consider them in revising their paper. Please consider them as positive feedback and note that I am not an expert in statistical methods.

Reviewer Comment #1: Title: The authors could reflect on the design of the study in the title itself besides mentioning the registry analysis.

Author Response: Thank you for the opportunity to strengthen our title. We have changed the title to “High levels of Post-Traumatic Stress Disorder symptoms among healthcare workers during the COVID-19 pandemic: An Analysis of the HERO registry.”

Reviewer Comment #2: Abstract: Please consider removing the exact values in odd ratios. Providing statements of your findings should suffice in the results.

Author Response: Thank you for the suggestion to restructure our abstract. We have removed the odds ratio from the abstract. 

Reviewer Comment #3: Introduction: I suggest the authors to give a brief overview on why they think COVID-19 can be equated with Trauma definition for PTSD before proceeding on to literature depicting relation between PTSD and COVID-19.

Author Response: We appreciate the opportunity to clarify this important point. We have added in the following text to the introduction.

“Post-traumatic stress disorder (PTSD) is a disorder that develops in some people who have experienced a shocking, scary, or dangerous event [1]. Common examples of harmful experiences that may result in PTSD include military combat, natural disasters, and violent assault. Recent data has shown that the chronic strain caused by pandemics has a similar psychological impact on a variety of health outcomes in HCWs, including burnout, depression, and anxiety [2].”

Reviewer Comment #4) The authors mention two objectives. When they say burden of PTSD symptoms, what do they mean? It is understandable that persistent stressor like COVID-19 will lead to an increase in levels of stress for most populations, including HCWs. How do they estimate that these symptoms are PTSD symptoms and not anxiety and/ or depression? When they say elevated level of PTSD symptoms, what is the baseline against which they measure the symptoms?

Author Response: We thank the author for this insightful comment. We agree that we are assessing two outcomes, including the prevalence of PTSD symptoms among health care workers in the HERO registry, and the association between HCW characteristics (race, gender, professional role, work setting, and geographic region). We understand the confusion with the use of term “burden” as this implies measures of impact. We have modified the text referencing “burden” of PTSD symptoms to focus on “prevalence”.

The survey used in the study was validated as a modified screening tool used for PTSD symptoms. As described in the manuscript, this tool is not used to formally diagnose PTSD or other mental health disorders, such as anxiety or depression, but rather identify potential symptoms associated with PTSD. We appreciate the opportunity to clarify and have added the following language to the methods section: “We used a previously validated PTSD questionnaire [17]”. We have additionally included the following statement in the limitations section:

“Furthermore, we did not measure information about prior PTSD diagnosis or treatment and used a brief scale to measure PTSD symptomatology. A more in-depth psychometric scale, including items about anxiety, depression, burnout along with PTSD would give us a greater idea of true PTSD scores.”

Although PTSD symptom prevalence was high, we did not have a baseline measurement that we were able to compare against. We appreciate the reviewer’s concern about this challenge, and this issue has been highlighted in our limitation section. Unfortunately, given the limitations of the study design, we could not assess for baseline PTSD symptoms in HCWs prior to the pandemic.

Reviewer Comment #5) Methods: What were the eligibility criteria except for being HCWs? Were there any duration criteria for symptoms mentioned?

Author Response: Thank you for the opportunity to clarify the inclusion/exclusion criteria. While the overall HERO Registry enrolled non-HCWs, most participants identified as HCWs, defined as working in a setting where people received healthcare. The study population for this analysis included only HCWs 18 years of age and older who were able to speak and read English or Spanish. There were no symptom duration criteria. We have clarified these criteria in the text of the methods section. 

Reviewer Comment #6) I understand this was a voluntary registry- based study. However, Regarding sampling…was there any justification for sample size? How many invitations were sent? What was the response rate? What about missing data? Were there any outliers? Also, mention on the design of the study.

Author Response: Since the study was voluntary and conducted entirely online, we did not set an enrollment cap. We have expanded our description of this in the paper as shown below and have included it as a limitation. 

“The Registry was a fully remote online study that captured perspectives through ongoing virtual surveys and return of results. Although participation in the HERO Registry was through an online portal, enrollment and engagement for the Registry supported by the PCORnet infrastructure and 34 PCORnet sites. The study primarily enrolled participants through marketing campaigns and word of mouth, but also through targeted recruitment efforts at PCORnet sites participating in one of the research projects leveraging the Registry (HERO-HCQ or HERO-TOGETHER) and that used the Registry for pre-screening for project eligibility. Participants self-enrolled and responded to surveys through an online portal that was supported by desktop and mobile applications.

Furthermore, all HERO participants who consented between 4/10/2020 and 5/6/2022 and left a valid email address were sent an invite through email and were given the opportunity to participate in the PTSD survey during the week between 4/29/2022 and 5/6/2022. By querying HERO data, 51,464 participants consented between 4/10/2020 and 5/6/2022, and 3,593 (7%) responded to the PTSD survey. This description has been added to the methods section. 

Reviewer Comment #7) The authors mentioned cohort in one instance consisting of HCWs, their family members and members of the community. What was the relevance of this and why is it termed as cohort? Please clarify.

Author Response: This was used as a synonym for “study population”. We have updated the term to “study population” for consistency. Additionally, we have added clarification language into the methods that describe although the overall Registry population consist of family members and members of the community, our analysis focused solely on the cohort of HCWs.

Reviewer Comment #8) Survey Tools: DSM- 5 PTSD screening survey: I suggest the authors to briefly mention regarding this instrument and list its psychometric properties in the given context. The questions in this survey appear vague and threshold of 3 could be met by any depressed or anxious individual as well. Please comment on this statement. 

Author Response: As described in the limitations section, this survey was not intended to represent formal PTSD diagnosis, but rather used as a screening survey to identify potential PTSD symptoms. We have clarified this further in the text. We appreciate the reviewers concern about specificity of the scale and have added the following qualifications into the limitations section, “A more in-depth psychometric scale, including items about anxiety, depression, burnout along with PTSD would give us a greater idea of true PTSD scores”. 

Reviewer Comment #9a) Statistical Analysis: I suggest the authors to comment on the distribution of the variables of interest. I see that median has been reported for age, and I assume that it is skewed. Please provide details regarding other variables. Also, please cite the reference for PTSD threshold as 3 and above. 

Author Response: Thank you for the opportunity to provide more detail about the statistical analysis. First, we have edited the tables to report median and interquartile range (25th – 75th percentile) for age. The distribution of age is also described using categorical age percentage. Although a PTSD score cutoff of 4 is used in general population, a lower cutoff has previously been proposed for female populations [19]. Because our sample is predominantly female (>80%), we chose a cutoff score of 3 to indicate PTSD symptoms. We have added this as an additional reference and text into the manuscript. 

b) Regarding unadjusted model, which statistical test was used? Were assumptions met for multivariate logistic regression? Please provide rationale for inclusion of variables in adjusted models.

The unadjusted model means only one explanatory variable (or the corresponding row variable in model table) was fitted in the model. We have updated the methods section describing the models. For logistic regression, we analyzed the binary endpoint of PTSD score of 3 or higher, and we checked for collinearity between explanatory variables using tolerance and VIF. We did not observe any collinearity issues and therefore proceeded with the logistic regression approach. Adjustment variables comprised demographics and HCW job characteristics collected in HERO surveys, and were chosen by the HERO team investigators based on substantive knowledge of key factors that could impact the PTSD outcome. 

Reviewer Comment #10) Discussion: Discussion needs rewriting. In regard to first objective, how do the authors conclude that there is high level of PTSD symptoms in all levels of HCWs. Please justify.

Author Response: We appreciate this suggestion. We have substantially revised to improve the clarity of this important point in the discussion to clarify that certain groups, such as nurses, had higher prevalence of PTSD symptoms. 

Reviewer Comment #11) In line 177- 178, the authors conclude with the statement that excessive stress and loss of resilience occurs to the HCWs during the pandemic. Please clarify on how raised PTSD symptoms means loss of resilience.

Author Response: We thank the author for the opportunity to clarify this statement. We have removed the term “loss of resilience”. 

Reviewer Comment #12) Discussion on the association between gender and PTSD is not enough since there is abundant literature on this. Most of the discussion appears generic on relation between psychological sequalae and gender. Please be more specific to in relation to PTSD. Same goes for association with nurses compared to physicians. With age, the evidence is mixed. Please discuss in light of those findings.

Author Response: We appreciate the opportunity to add more information to improve our discussion section. We have made 3 major changes in response to the reviewer’s comment.

First, we have included additional citations related to gender and PTSD that are consistent with our results. Our findings corroborate the past literature that shows females (especially HCWs) are at higher rates for developing mental health disorders such as PTSD. 

Second, we have added citations to demonstrate consistency between our findings and those from previous studies demonstrating that compared to physicians, nurses during pandemics may be at higher risk for developing PTSD symptoms. 

Third, we have included additional detail related to our findings of an association between PTSD symptoms and age. Our findings are congruent with past literature demonstrating the association between combat PTSD and age, however, we recognize that more research needs to be done to further understand the relationship between pandemic driven PTSD symptoms and age. 

Reviewer Comment #13) Limitations: The authors have listed few limitations. I believe this study has more limitations and biases such as using only screening questionnaire, not ruling out depression or anxiety or even burnout, which are quite common in this scenario. There could be many other factors at play which have not been considered in the given study.

Author Response: Thank you for the opportunity to add greater detail to the limitation section. We agree that there are limitations of our study design. We have expanded the limitations section to include acknowledgment that our scale was designed to measure PTSD symptoms only and did not include questions on depression, anxiety, and burnout. Despite these limitations, we believe our study provides an important starting point for more comprehensive investigation of the mental health implications of the COVID-19 pandemic in HCWs, particularly those related to trauma. 

Reviewer Comment #14) I also suggest the authors to highlight the strengths of this study despite the limitations. 

Author Response: Thank you for the suggestion. While there are limitations with our study, we did analyze the rate of PTSD symptoms of large cohort of HCWs during the COVID-19 pandemic. We appreciate the opportunity to add additional information that highlight the strengths of the study. 

Reviewer Comment #15) I wish the authors all the best.

Author Response: We appreciate the thoughtful and considerate feedback. With the following revisions in place, our paper has increased depth and clarity. We are thankful for the reviewer’s input. 

Additional Reviewer Comment: The PLOS Data policy requires authors to make all data underlying the findings described in their manuscript fully available without restriction, with rare exception (please refer to the Data Availability Statement in the manuscript PDF file). The data should be provided as part of the manuscript or its supporting information, or deposited to a public repository. For example, in addition to summary statistics, the data points behind means, medians and variance measures should be available. If there are restrictions on publicly sharing data—e.g. participant privacy or use of data from a third party—those must be specified.

Author Response: We are in the process of depositing the data with PCORI right now, so it will be publicly available by the end of the year.

---

## [Decision Letter · Decision Letter 1]

24 Sep 2023

PONE-D-23-06322R1High levels of Post-Traumatic Stress Disorder symptoms among healthcare workers during the COVID-19 pandemic: A case control analysis of the HERO registryPLOS ONE

Dear Dr. Rice,

Thank you for submitting your manuscript to PLOS ONE. After careful consideration, we feel that it has merit but does not fully meet PLOS ONE’s publication criteria as it currently stands. Therefore, we invite you to submit a revised version of the manuscript that addresses the points raised during the review process.

We look forward to receiving your revised manuscript.

Kind regards,

Vincenzo De Luca

Academic Editor

PLOS ONE

Journal Requirements:

Reviewers' comments:

Reviewer's Responses to Questions

**Comments to the Author**

1. If the authors have adequately addressed your comments raised in a previous round of review and you feel that this manuscript is now acceptable for publication, you may indicate that here to bypass the “Comments to the Author” section, enter your conflict of interest statement in the “Confidential to Editor” section, and submit your "Accept" recommendation.

Reviewer #1: All comments have been addressed

2. Is the manuscript technically sound, and do the data support the conclusions?

Reviewer #1: Yes

3. Has the statistical analysis been performed appropriately and rigorously? 

Reviewer #1: Yes

4. Have the authors made all data underlying the findings in their manuscript fully available?

Reviewer #1: No

5. Is the manuscript presented in an intelligible fashion and written in standard English?

Reviewer #1: Yes

6. Review Comments to the Author

Reviewer #1: 22/08/2023

Thank you again for the opportunity to review the revised manuscript.

The authors have done good job in reviewing the manuscript by addressing most comments. Overall, the comments are satisfactory. I have few concerns now as follows.

Comment number 3 in the previous revision is not satisfactory. How can COVID-19 be equated with trauma in accordance with ICD or DSM? Please be more concrete in justification.

In the manuscript, the authors mention that >80% of the samples were females and in results, the nurses have more PTSD symptoms. So, could it be because of the select sample? Please clarify.

Otherwise, the manuscript is good to go.

All the best.

7. PLOS authors have the option to publish the peer review history of their article (what does this mean?). If published, this will include your full peer review and any attached files.

Reviewer #1: **Yes: **Saraswati Dhungana

---

## [Author Response · Author response to Decision Letter 1]

29 Sep 2023

September 29th, 2023 

PONE-D-23-06322R1

Post-Traumatic Stress Disorder symptoms among healthcare workers during the COVID-19 pandemic: Analysis of the HERO registry

Dear Vincenzo De Luca,

Thank you very much for the supportive review of our manuscript. We appreciate the additional helpful suggestions from the editor and reviewers and have made the following modifications to improve the presentation of our analysis. We believe the manuscript is greatly improved as a result of these changes.

Again, we appreciate PLOS ONE’s support in the investigation of the importance of PTSD symptoms among healthcare workers during the COVID-19 Pandemic and believe this work will be of great interest to your readership. 

All reviewer comments are listed in bold. Please find our responses below the reviewer’s comment.

Sincerely,

Emily O’Brien, PhD, FAHA

Associate Professor 

Duke Clinical Research Institute

Duke University School of Medicine

Department of Population Health Sciences

215 Morris Street, Suite 210

Durham, NC 27701 

REVIEWER #1 Comment:

Thank you again for the opportunity to review the revised manuscript.

The authors have done good job in reviewing the manuscript by addressing most comments. Overall, the comments are satisfactory. I have few concerns now as follows.

Comment number 3 in the previous revision is not satisfactory. How can COVID-19 be equated with trauma in accordance with ICD or DSM? Please be more concrete in justification.

In the manuscript, the authors mention that >80% of the samples were females and in results, the nurses have more PTSD symptoms. So, could it be because of the select sample? Please clarify.

Otherwise, the manuscript is good to go.

All the best.

Author Response: Thank you for the opportunity to clarify this first point and add depth to our introduction. Per the reviewer’s suggestion, we have added the following languages and references. 

“Prior research indicates that ‘helpers’ during human disasters often face psychological trauma (1). Analyses of post-9/11 experiences among first responders underscored the psychological risks for professionals exposed to suffering (2), with terms like compassion fatigue, secondary traumatic stress, and vicarious trauma used to describe such effects. This suggests that healthcare providers treating COVID-19 patients may also experience similar trauma, given their critical role in response to this crisis.”

Furthermore, we understand the reviewer's concern that our sample size was largely female and therefore there might be some confounding by gender and professional role. Our adjusted model should account for confounds such as gender but we recognize that can’t rule out residual confounds by other unmeasured factors. We have added the following language to the limitation section to address this. 

“Moreover, although our adjusted model controlled for several important confounders such as gender, we cannot rule out the possibility of residual confounding from additional unmeasured factors.”

---

## [Decision Letter · Decision Letter 2]

12 Oct 2023

High levels of Post-Traumatic Stress Disorder symptoms among healthcare workers during the COVID

-19 pandemic: A case control analysis of the HERO registry

PONE-D-23-06322R2

Dear Dr. Rice,

We’re pleased to inform you that your manuscript has been judged scientifically suitable for publication and will be formally accepted for publication once it meets all outstanding technical requirements.

Kind regards,

Vincenzo De Luca

Academic Editor

PLOS ONE

Additional Editor Comments (optional):

Reviewers' comments:

Reviewer's Responses to Questions

**Comments to the Author**

1. If the authors have adequately addressed your comments raised in a previous round of review and you feel that this manuscript is now acceptable for publication, you may indicate that here to bypass the “Comments to the Author” section, enter your conflict of interest statement in the “Confidential to Editor” section, and submit your "Accept" recommendation.

Reviewer #1: All comments have been addressed

2. Is the manuscript technically sound, and do the data support the conclusions?

Reviewer #1: Yes

3. Has the statistical analysis been performed appropriately and rigorously? 

Reviewer #1: Yes

4. Have the authors made all data underlying the findings in their manuscript fully available?

Reviewer #1: Yes

5. Is the manuscript presented in an intelligible fashion and written in standard English?

Reviewer #1: Yes

6. Review Comments to the Author

Reviewer #1: (No Response)

7. PLOS authors have the option to publish the peer review history of their article (what does this mean?). If published, this will include your full peer review and any attached files.

Reviewer #1: **Yes: **Saraswati Dhungana

---

## [Editor Report · Acceptance letter]

31 Oct 2023

PONE-D-23-06322R2 

Post-Traumatic Stress Disorder symptoms among healthcare workers during the COVID-19 pandemic: Analysis of the HERO Registry 

Dear Dr. Rice:

I'm pleased to inform you that your manuscript has been deemed suitable for publication in PLOS ONE. Congratulations! Your manuscript is now with our production department. 

Kind regards, 

on behalf of

Dr. Vincenzo De Luca 

Academic Editor

PLOS ONE